# Colonization and Authentication of the Pyrethroid-Resistant *Anopheles* *gambiae* s.s. Muleba-Kis Strain; an Important Test System for Laboratory Screening of New Insecticides

**DOI:** 10.3390/insects12080710

**Published:** 2021-08-08

**Authors:** Salum Azizi, Janneke Snetselaar, Alexandra Wright, Johnson Matowo, Boniface Shirima, Robert Kaaya, Rashid Athumani, Filemoni Tenu, Natacha Protopopoff, Matthew Kirby

**Affiliations:** 1Department of Medical Parasitology and Entomology, Kilimanjaro Christian Medical University College, Moshi 255, Tanzania; janneke.snetselaar@lshtm.ac.uk (J.S.); johnson.matowo@pamverc.or.tz (J.M.); boniface.shirima@pamverc.or.tz (B.S.); robert.kaaya@pamverc.or.tz (R.K.); rashid.athumani@pamverc.or.tz (R.A.); filemoni.tenu@pamverc.or.tz (F.T.); Natacha.Protopopoff@lshtm.ac.uk (N.P.); matthew.kirby@lshtm.ac.uk (M.K.); 2Pan African Malaria Vector Research Consortium, Moshi 255, Tanzania; 3The Innovative Vector Control Consortium, Liverpool L3 5QA, UK; 4Department of Disease Control, London School of Hygiene and Tropical Medicine, London WC1E 7HT, UK; Lauren.wright@lshtm.ac.uk

**Keywords:** insecticide selection, out-crossing, strain authentication, laboratory screening

## Abstract

**Simple Summary:**

Malaria control and prevention have traditionally relied on the use of insecticides in the form of treated bed nets or residual spraying in households. However, scaling up of these interventions—based on few available insecticide classes—resulted in the development and spread of insecticide resistance in malaria-transmitting mosquitoes. There is therefore an urgent need for introducing and applying new insecticides that are effective against these mosquitoes. Laboratories tasked with evaluating the efficacy of novel insecticides need to establish a large colony of resistant mosquitoes. In this study, we report the procedures used and challenges faced during the establishment and maintenance of a resistant mosquito strain in the laboratory which reflects the characteristics of the wild-resistant mosquito populations found in East Africa.

**Abstract:**

Background: The emergence and spread of insecticide resistance in malaria vectors to major classes of insecticides call for urgent innovation and application of insecticides with novel modes of action. When evaluating new insecticides for public health, potential candidates need to be screened against both susceptible and resistant mosquitoes to determine efficacy and to identify potential cross-resistance to insecticides currently used for mosquito control. The challenges and lessons learned from establishing, maintaining, and authenticating the pyrethroid-resistant *An. gambiae* s.s. Muleba-Kis strain at the KCMUCo-PAMVERC Test Facility are described in this paper. Methods: Male mosquitoes from the F_1_ generation of wild-pyrethroid resistant mosquitoes were cross-bred with susceptible female *An. gambiae* s.s. Kisumu laboratory strain followed by larval selection using a pyrethroid insecticide solution. Periodic screening for phenotypic and genotypic resistance was done. WHO susceptibility tests and bottle bioassays were used to assess the phenotypic resistance, while Taqman™ assays were used to screen for known target-site resistance alleles (*kdr* and *ace-1*). Additionally, the strains were periodically assessed for quality control by monitoring adult weight and wing length. Results: By out-crossing the wild mosquitoes with an established lab strain, a successful resistant insectary colony was established. Intermittent selection pressure using alphacypermethrin has maintained high *kdr* mutation (leucine-serine) frequencies in the selected colony. There was consistency in the wing length and weight measurements from the year 2016 to 2020, with the exception that one out of four years was significantly different. Mean annual wing length varied between 0.0142–0.0028 mm compared to values obtained in 2016, except in 2019 where it varied by 0.0901 mm. Weight only varied by approximately 0.001 g across four years, except in 2017 where it differed by 0.005 g. Routine phenotypic characterization on Muleba-Kis against pyrethroids using the WHO susceptibility test indicated high susceptibility when type I pyrethroids were used compared to type II pyrethroids. Dynamics on susceptibility status also depended on the lapse time when the selection was last done. Conclusions: This study described the procedure for introducing, colonizing, and maintaining a resistant *An. gambiae* s.s. strain in the laboratory with leucine to serine substitution *kdr* allele which reflects the features of the wild-resistant population in East Africa. Challenges in colonizing a wild-resistant mosquito strain were overcome by out-crossing between mosquito strains of desired traits followed by intermittent insecticide selection at the larval stage to select for the resistant phenotype.

## 1. Introduction

Malaria vector control principally relies on the use of Insecticidal Treated Nets (ITNs) and Indoor Residual Spraying (IRS) as the most effective measures to prevent malaria transmission [1]. Historically, pyrethroids were used extensively for conventionally treated nets, superseded by Long Lasting Insecticidal Nets (LLINs), and also used for IRS due to their efficacy, relatively long persistence compared to other insecticides [2,3,4], and perceived low toxicity to humans [5,6,7,8]. However, the development and spread of pyrethroid resistance in malaria vector populations [4,9] demanded the development of new classes of insecticides with novel modes of action (MoA) for the control of mosquitoes and other disease vectors [10,11,12,13].

In developing new insecticides, several stakeholders are required in the process. The Innovative Vector Control Consortium (IVCC) has pioneered the bonding of prime agrochemical innovator industries, with research and academic institutions as key stakeholders in developing and evaluating new insecticides for mosquito control to prevent malaria and other neglected tropical diseases [14]. Research institutions perform laboratory and field screening of new chemistries for efficacy against mosquito populations and identify any cross-resistance risks at an early stage in the product development pipeline [15]; in this process mosquitoes are required as test systems [15,16]. In response to the global escalation of insecticide resistance in mosquito vectors, the WHO specifically recommends the establishment, authentication, and use of resistant mosquito strains during phase I efficacy testing of new non-pyrethroid insecticides [16]. This recommendation ensures that the evaluation will be able to capture efficacy against current resistance in malaria vectors. Authentication of a new insectary strain involves routine confirmation of the unique characteristics of the strain that sufficiently distinguish it from all others held in the same facility. This comprises routine validation of the species or subspecies identity, plus the resistance status as defined by genotypic and/or phenotypic characteristics [17,18]. In establishing a resistant insectary colony under artificial rearing conditions, the field sourced mosquitoes undergo several bottlenecks that could impair its suitability for the tests. Due to lack of variation and complexity in artificial rearing conditions, adaptation to these settings can favor populations to evolve in new directions from wild populations, especially when selection pressures and nutrition differ between the two settings [19,20]. Laboratory maintenance of insects in discrete generations facilitates selection for individuals that reproduce early and develop faster [21,22]. It is reported that adaptation to artificial environment can result in significant rapid evolutionary traits changing compared to natural populations [23,24]. This can lead to problems when reared insects are intended for release as biocontrol agents or in sterile insect control programs, when using laboratory strains to comprehend field population dynamics, and when using reared strains to predict vector control tools’ effectiveness in the field. Attempts have been made to minimize the genetic drift and inbreeding effects through crossing an established laboratory stock with outbred field stock [25,26]. However, there is less utility for crossing the laboratory strain with field mosquitoes to maintain a complete genetic background of field populations when the colony is established to serve as a close representative for a few defined traits which can be fixed, and when the ultimate use is limited to laboratory and semi-field environments.

In 2008, the Insecticide Testing Facility (ITF) of the Kilimanjaro Christian Medical University College-Pan-African Malaria Vector Research Consortium (KCMUCo-PAMVERC) Test Facility in Moshi Tanzania was initiated in parallel with a molecular laboratory, two insectaries, and three field stations. In the insectaries, the Test Facility established *Aedes*, *Anopheles,* and *Culex* mosquito colonies of different insecticide resistance profiles. From 2008–2011, *Anopheles* mosquitoes kept at the KCMUCo-PAMVERC Test Facility were limited to susceptible *An. gambiae* sensu stricto Kisumu (susceptible to all classes of insecticides used for vector control) and *An. arabiensis* collected from lower Moshi and reared to first filial generation (F1), the pyrethroid-resistant vector local to the Test Facility [27,28]. In 2012, the Test Facility acted to establish a colony of pyrethroid-resistant *An. gambiae* s.s. that would represent a typical East African resistant population. The *An. gambiae* Muleba-Kis strain was established and has been maintained in the insectary for years and propagated over hundreds of generations successfully, a feature emphasized by some scientists to qualify a colony as a strain [29]. The established *An. gambiae* Muleba-Kis strain is similar to East African *An. gambiae* s.s. populations for having the East African knockdown resistance (L1014S), a sodium channel mutation in *An. gambiae* that confers DDT and pyrethroid resistance [30]. The origin of L1014S mutation is Eastern Africa [30,31], hence the name *kdr*-east, although currently this mutation is no longer geographically restricted to East Africa [32,33] and its occurrence is frequently associated with the West African mutation L1014F [33,34]. Different types of pyrethoids, namely type I and type II, affect mosquitoes with *kdr* (East or West or mixture) differently. Pyrethroids are classified based on their chemical structures; type I pyrethroids lack the cyano-moiety present at the α position of type II pyrethroids. The type II pyrethroids generally delay the inactivation of the voltage-gated sodium channel substantially longer, and their effects are less reversible than type I pyrethroids [35]. Due to the similar steric profile with pyrethroids, DDT, an organochlorine, is affected with resistance to pyrethroids which often provides cross-resistance to DDT. A study by Reimer reported that mosquito populations carrying a high *kdr* frequency showed more resistance to DDT and type I pyrethroids than to type II pyrethroids [36].

The L1014S mutation has been fixed in a population of *An. gambiae* s.s. in Muleba District, north-western Tanzania [37,38], Busia, and Mayuge Districts in Eastern Uganda [39]. The occurrence of the L1014S mutation but at lower frequencies has been reported elsewhere in Tanzania [40], Kenya [41], and Uganda [42,43]. In previous studies done in Muleba district, where mosquitoes for this study were collected, it has also been reported that mosquitoes are resistant to bendiocarb, DDT, permethrin and deltamethrin, although there was no evidence for *Ace*-1 mutation [37]. Another study, a national-wide survey for resistance [44], reported *An. gambiae* s.l. resistance to pirimiphos-methyl for the first time in three sites (including Muleba district) out of 20 sites in Tanzania. Since the target site to organophosphates and carbamates is the AChE enzyme and that resistance in mosquitoes to this target site is frequently a G119S mutation in the *ace-1* gene [45], it is therefore reasonable to characterize L1014S and Ace-1 mutations as desired traits in the established colony to resemble the parental resistant population.

In this paper, we describe the procedures undertaken at the KCMUCo-PAMVERC Test Facility to establish a pyrethroid-resistant strain called *An. gambiae* s.s. Muleba-Kis. Here we focus on the procedures and lessons learned from out-crossing, artificial resistance selection, bioassays, and genotyping assays used to authenticate this strain for over two hundred generations. We describe data on the stability of resistance traits and fitness parameters over eight years. These data provide baseline resistance information on the outcome of the long-term intermittent selection of mosquito larvae.

## 2. Materials and Methods

### 2.1. Study Site 

From April to May 2012, *An. gambiae* s.l. mosquitoes were collected in houses in two villages: Kyamyorwa (02°04′27.5″ S, 31°34′10.8″ E) and Kiteme (02°03′20.9″ S, 31°27′16.8″ E) in Muleba, a rural district on the western shore of Lake Victoria in northwest Tanzania (Figure 1).

The mosquito collection for this study was part of an ongoing large cluster randomized trial in Muleba district, north-western Tanzania [46]. *An. gambiae* s.s. were the main vectors found in this area and have historically exhibited high resistance levels to pyrethroids [28], with mortality after exposure not exceeding 35%. The L1014S point mutation associated with pyrethroid resistance was nearly fixed, while no *Ace*-1 mutation was found [37].

### 2.2. Timeline

The timeline below, Figure 2, indicates the sequence of activities in this study across generations of *An. gambiae* Muleba-Kis.

### 2.3. Collection of Wild Mosquitoes and Introduction into the Insectary 

Indoor resting blood-fed *Anopheles* were collected in house bedrooms using mouth aspirators. Mosquitoes were transferred in paper cups supplemented with glucose and transported to field insectaries located in Muleba. They were held under ambient relative humidity and temperature conditions in 30 × 30 × 30 mosquito cages containing a petri dish of moistened cotton wool overlaid with damp filter paper for egg laying. After laying, adult *An. gambiae* s.l. were stored individually and subsequently identified by Polymerase Chain Reaction (PCR) [47]. Collections were done over two months and eggs (approximately 500 eggs) were sent to the KCMUCo-PAMVERC Test Facility. Eggs were introduced into plastic bowls (6 L capacity) filled with 4 L of water. Larvae were reared under ambient temperature and relative humidity and fed with cereal for infants (Cerelac^®^, Nestlé Kenya Limited, Pate, Kenya) mixed with ground sardines at a 2:1 ratio. Adult mosquitoes were reared at 60–90% RH and 20–35 °C in cages (30 cm × 30 cm × 30 cm) covered with untreated netting material and provided with glucose solution 10%. To ensure optimal rearing conditions, insectary larval density was restricted to 200–300 per bowl (3 L capacity), water for mosquito rearing was pre-boiled to avoid bacterial infections, and environmental conditions (water and air temperature, relative humidity) were monitored and maintained.

### 2.4. Crossing for “Insectary Vigor”

When F1 mosquitoes were five days old, a restrained guinea pig was introduced into the cages of mosquitoes that were starved for one hour prior to blood-feeding. To overcome difficulties of adaptation to insectary conditions, out-crossing between female *An. gambiae* Kisumu and male *An. gambiae* Muleba mosquitoes were conducted. The main difficulties encountered were low blood-feeding, egg-laying, and survival, otherwise known as “insectary vigor.” The *An. gambiae* Kisumu strain was obtained in 2008 through BEI Resources, NIAID, NIH: *Anopheles gambiae*, Strain KISUMU1, Eggs, MRA-762, contributed by Vincent Corbel. This strain is originating from Kisumu, Kenya, and was successfully established at our insectary and feeds well on guinea pigs. The Muleba and Kisumu strain pupae were collected separately, and males were separated from females on the first day after emerging. Adult male Muleba and female Kisumu mosquitoes were mixed at a ratio of 50:50 in a mosquito cage. These were reared at 20–35 °C, 60–90% RH, and a natural 12:12 h L:D photoperiod, and were provided with a guinea pig for blood-feeding and filter paper medium for egg-laying. This successful outcrossed mosquito was then named “*An. gambiae* s.s. Muleba-Kis strain” and has been reared at the KCMUCo-PAMVERC test facility since 2013.

### 2.5. Selection to Maintain Pyrethroid Resistance

In this study, selection was based on the exposure of larval mosquitoes to pyrethroid insecticides, and pyrethroids were chosen due to intensive usage in public health and having the most widespread resistance among mosquito vectors across Africa [48,49]. Artificial selection for pyrethroid resistance was started in the 15th generation. Six bowls each with around 100 larvae of 3rd to 4th instars were used initially, adopting a modified method by Shidrawi [50]. One mL of insecticide solution was added to 1 L of tap water at 27–32 °C, stirred for 1 min using a Pasteur pipette, and then left for 10 min to allow evaporation of acetone which was used as a solvent for insecticide solution preparation. Larvae were transferred into the glass bowl with the dissolved insecticide solution, each bowl with around 100 larvae. A small amount of larvae food was added and larvae were left for 24 h in the selection bowl. After 24 h, mortality was estimated. Mortality was estimated in three categories: high mortality, 67–100%; moderate mortality, 34–66%; or low mortality, 0–33%. The initial selection was done using permethrin, and later alphacypermethrin was used for colony selection. The initial permethrin concentration used for the section was 0.1 mg/L and increased to 0.2 mg/L at a time when larvae mortality was in a low category. The initial alphacypermethrin concentration was 0.025 mg/L and it increased to 0.05 mg/L when larvae mortality was in a low category. The larvae were sieved when still alive from the selection bowl, rinsed with 500 mL water (temp 27–32 °C) and returned to their original six bowls, and reared, while the dead larvae were removed. The selection was conducted intermittently. The availability of technical grade insecticide to make up the selection solutions and a need for mosquitoes for ongoing laboratory bioassays were the main constraints preventing routine artificial section of the colony.

### 2.6. Authentication of the Outcrossed An. Gambiae s.s. Muleba-Kis Strain 

#### 2.6.1. Phenotypic Resistance

##### WHO Susceptibility Test and CDC Bioassay

Insecticide susceptibility bioassays were done from the 17th to 196th generation, in accordance with WHO guidelines [51]. Bioassays were carried out using six insecticides, namely permethrin (0.75%), alphacypermethrin (0.05%), deltamethrin (0.05%), DDT (4%), bendiocarb (0.1%), and pirimiphos-methyl (0.25%), and tests were conducted at 25 ± 2 °C and 80 ± 10% relative humidity. Each type of insecticide bioassay was performed in 5 replicates, including one as a control. Twenty to 25, two-to-five-day-old female, blood unfed mosquitoes were tested, constituting a sample size of 100 to 125 mosquitoes for each insecticide. Tested mosquitoes were monitored for knockdown at 60 min and mortality at 24 h post exposure. In parallel with permethrin papers, limited WHO susceptibility bioassays were also conducted against bendiocarb papers (0.1%) and pirimiphos-methyl (0.25%). The insecticide resistance of the selected colony at the 190th generation was compared to the susceptible Kisumu strain using α-cypermethrin in CDC bottle bioassay [52] at concentrations of 52.5, 25.7, 12.5, 6.1, 3, 1.5, and 0 µg/bottle, where 12.5 µg/bottle acted as a discriminating concentration for *Anopheles* [51].

##### Synergist-Insecticide Bottle Bioassay

In a separate experiment (unpublished) in 2018, at the 143rd generation, a synergist assay with piperonyl butoxide (PBO) was undertaken to assess the role of elevated mixed-function oxidases in resistance. One hundred, 2–5-day-old female An. gambiae Muleba-Kis mosquitoes were tested for metabolic resistance using Piperonyl butoxide (PBO) in four replicates (25 mosquitoes per bottle) at a concentration of 100 µg/mL for one hour pre-exposure and then followed by 30 min exposure to permethrin (21.5 µg/mL), in accordance to the CDC guidelines [52], with the exception that mortality was considered at 24 h post-exposure. In brief, mosquitoes were pre-exposed to either acetone-coated bottles or PBO for 1 h at a temperature and humidity of 27 ± 2 °C and 70 ± 10% RH, respectively, during and after exposure. After pre-exposure, mosquitoes were transferred to holding cages for 60 min before being exposed for 30 min to bottles coated with either 21.5 µg/mL of permethrin or acetone as a control. After exposure, the mosquitos were transferred into holding cups and provided with 10% glucose-soaked cotton pads. After 60 min, post-exposure knockdown was recorded, and mortality was recorded 24 h post-exposure.

#### 2.6.2. Genotypic Basis of Resistance

##### Detection of *kdr* and *Ace*-1 Mutations

The frequency of leucine to serine mutations (L1014S), termed *kdr* east (*kdr*-e) and *ace*-1, were assessed and frequently monitored using the method described by Bass et al. [47,53] to monitor progress in resistance development after successive insecticide selection events. For *Ace*-1 and/or *kdr*-e alleles, a total of 84–88 samples were analyzed by PCR per each test.

#### 2.6.3. Species Identification and Biometric Measures for Fitness

##### Species Identification

To ensure colony species purity, at 43, 99, 131, 150, 162, 168, 178, 188, 190, 198, and 204th generations, the PCR for species identification was conducted using single nucleotide polymorphism genotyping [47]. At each generation, a total of 84-88 mosquito samples were tested.

##### Biometric Measurements

The size of individual adult females was estimated by the average length of left wings, while weight was measured by weighing the whole mosquito. To measure the wing length, a total of one hundred Muleba-Kisumu females were randomly sampled from five selected mosquito-rearing cages quarterly, covering the 99th to 204th mosquito generations. The wings were cut and placed on a stage micrometer (10 mm long with 100 × 0.1 mm (100 μm) divisions). Wing length was measured as the distance from the alula to the end of the wing where vein three ends [54,55,56] using an ocular micrometer at 2X objective magnification on a Nikon stereomicroscope, Model; SMZ 645 [Nikon Instruments, 1300 Walt Whitman Road, Melville, NY 11747-3064, U.S.A.], see Figure 3 below.

Wing length and weight were continuously monitored in succeeding years regardless of whether selection with insecticide selection was done or not.

### 2.7. Statistical Analysis

The WHO criteria were used to classify the resistance or susceptibility status of the tested mosquito populations [51]. Descriptive analysis was performed to check for normality on wing and weight measures from the samples. Wing length and mosquito weight measures were all normally distributed. Using Stata [57], two sample T-tests were performed to compare wing length or mosquito weight across the years for the *An. gambiae* Muleba-Kisumu, and differences in mortality between *An. gambiae* Kisumu and *An. gambiae* Muleba-Kisumu across different concentrations in the CDC bottle bioassay. 

## 3. Results

### 3.1. Colony Selection

Progressive selection with permethrin from the 15th to 29th generations for Muleba-Kisumu strain caused a drastic drop in susceptibility, indicated by the decrease in mortality (Figure 4). Inexplicably, although the same insecticide type and concentration was used for selection, susceptibility increased from the 30th generation to 35th generation. From the 35th generation, selection was performed using alphacypermethrin. However, the selection with alphacypermethrin was not associated with an abrupt decrease of susceptibility, as selection was infrequent. 

### 3.2. Phenotypic Resistance

#### 3.2.1. WHO Susceptibility

The mortality observed in adult Muleba Kis exposed to permethrin (0.75%) test papers in the WHO susceptibility test was high (91% mortality) in the 17th generation (G17) and decreased to less than 20% at G25, then increased at G35. This follows a similar trend to the larvae mortality during selection procedures. The larvae selection with permethrin (pyrethroid types I) was not associated with a reduction in adult susceptibility when exposed to alpha-cypermethrin and deltamethrin (pyrethroid type II insecticides) using the WHO susceptibility bioassay at G35. Resistance to permethrin was the highest compared to the two other pyrethroids (α-cypermethrin and δ-methrin) from the 35th to 125th generations. In parallel with permethrin papers, WHO susceptibility bioassays conducted against bendiocarb papers (0.1%) and pirimiphos methyl (0.25%) resulted in 100% mortality, indicating that Muleba-Kis is fully susceptible to these insecticides. Mortality to DDT was consistently below 89% (Figure 5).

On average, the Muleba-Kisumu strain’s mortality was below the cutoff point (90%) when tested against permethrin (type I pyrethroid) and DDT papers. Only results with control mortality that were less than 20% were considered for analysis; tests when control mortality was higher than 20% were rejected. When tested against alpha-cypermethrin and deltamethrin (type II pyrethroids), the resistance level was low and above the cutoff value, which is suggestive of susceptibility to this pyrethroid class. However, although several mosquito mortalities were above 90%, during the 35th, 89th, 97th, and 125th generations mortalities scored below 98%, which could imply existence of resistance.

#### 3.2.2. Synergist-Insecticide Bottle Bioassay 

CDC bottle bioassays were conducted with permethrin (PRM) and piperonyl butoxide (PBO) against a susceptible strain and a resistant strain. The susceptible strain showed >98% knockdown and mortality after exposure to permethrin, both with and without pre-exposure to PBO. Muleba-Kis showed resistance to PRM (73% mortality), which was restored to susceptible levels (94% mortality) after pre-exposure to PBO, indicating likely involvement of metabolic resistance mechanism in the *An. gambiae* Muleba-Kis strain; see Figure 6 below.

### 3.3. Polymerase Chain Reaction (PCR) for Species Identification and Resistance Status

The *kdr* L1014S allele reached fixation in *An. gambiae* s.s. Muleba-Kis populations, coincident with the insecticide selection (Table 1). 

### 3.4. Resistance Strength of the Selected Colony: CDC Bottle Bioassay

Results from the CDC Bottle bioassay indicate that *An. gambiae* Muleba-Kisumu mosquitoes have lower mortality than *An. gambiae* Kisumu (Figure 7), which is suggestive of a higher level of pyrethroid resistance in the strain. 

At one and two times the diagnostic concentration of alphacypermethrin—12.5 µg/bottle and 25 µg/bottle, respectively—the Muleba-Kis strain showed significantly higher mortality than the Kisumu strain (two-sample *t*-test, *p* < 0.001). At four times the diagnostic concentration of the same insecticide—52.5 µg/bottle—there was no significant difference in mortality between the two strains.

Exposure of the Kisumu strain against alphacypermethrin in CDC bottles resulted in high mortality, indicating susceptibility against all doses, starting with a low dosage of 1.466 µg/bottle to the highest at 52.5 µg/bottle. On the other hand, exposure to the Muleba-Kis strain showed a dose-response, with mortality as low as 37% against the lowest dose and increasing to 98% mortality at four times the diagnostic dose.

### 3.5. Biometric Measures for Fitness 

A total of 450 mosquitoes were analyzed: 50 in 2016, 150 in 2017, 150 in 2019, and 100 in 2020. 

Data for female mosquito weight and wing length were normally distributed, hence we used the two-sample T-test to compare results between consecutive years. These results indicated that mosquito mean weight in 2017 was significantly higher than all other years, while the other years were similar to each other (Table 2).

On the other hand, mean wing length was only significantly higher in 2019 compared to the other years (Table 3).

## 4. Discussion

### 4.1. Blood-Feeding Challenges with Wild Mosquitoes

The propensity to feed on guinea pigs was not innate to the wild mosquito population and the colony could not be maintained by other means. The tendency to blood feed on guinea pigs was introduced by out-crossing, which is evidence for the genetic basis of intrinsic host-seeking factors within this Muleba mosquito strain. Host-seeking behaviors drive host choice, which is in turn driven by adaptive advantages that result from feeding on certain host species [58,59,60]. Wild mosquitoes were collected from bedrooms, which could indicate a preference of these mosquitoes to human blood. Similarly, observations from other studies [61,62] have associated host preference with the availability of host species for blood-feeding, which by their abundance form a readily accessible source of blood. This plasticity in host choice within mosquitoes could also be species- or strain-specific, accounting for differences in adopting a particular host as a blood source between different mosquito species or strains, as observed in this study where wild mosquitoes had a low affinity to guinea pig blood compared to the insectary-reared Kisumu strain.

### 4.2. Initial Low Insecticide Resistance Following Cross-Breeding

A common method used to establish resistant mosquito strain in the insectaries involves collecting wild-resistant mosquitoes and carefully maintaining them as they adapt to insectary conditions, usually going through a narrow bottleneck of few survivors in the first few generations. However, this endeavor has its challenges, such as failure of the wild strain to adapt to insectary temperature, relative humidity, and food; reduced mating; difficulties in blood-feeding on a new blood source; and reduced insecticide resistance. Early generations (15th to 17th) of Muleba-Kis strain in this study exhibited a low level of phenotypic resistance, which could be attributed to the low frequency of resistant alleles inherited from the resistant parent, the Muleba strain. The observed low frequency of resistant alleles, due to standing variation originating from the parental line before pesticide selection, is a phenomenon reported in other studies [63].

### 4.3. Impact of Mosquito Developmental Stage Used for Selection

The selection at the larval stage was chosen for three reasons. First, evolutionary pressure is strongest in young individuals to increase the probability of survival to reproductive maturity. Second, beneficial mutations at an older age can be associated with harmful effects in young individuals [64,65,66]. Third, by exerting the selection pressure to the aquatic stage of the mosquitoes, there is assurance for successive selection as it is impossible for larvae to survive subsequent selections but only through developing resistance [67]. Additionally, many reports have associated larvae exposure to trace amounts of pesticides with the development of insecticide resistance in malaria vectors [68,69,70,71].

In another study where larvae were selected, Shidrawi observed an increase of seven-fold resistance in an *Aedes* strain with initial moderate resistance when it was selected with DDT for eight generations [50]. When Shidrawi used different insecticides for the same strain over a different selection period, he obtained a different resistance outcome. On the other hand, in a study where adult *Anopheles* were selected [72], using a pyrethroid type II in a period of a single generation the mortality level decreased from 42% to 18% over one generation, reflecting an approximately two-fold increase in resistance. Although these results indicate that adult selection induces a more appreciable increase in resistance over a short period when compared to the larval selection, further research is needed to correlate the two stages using the same strain of mosquito and the same insecticide. Additionally, since selection in this study used different insecticides in different generations, it is difficult to determine the period without selection which is taken to reverse resistance to full susceptibility.

### 4.4. Impact of Selection Using Pyrethroids

The resistance of the Muleba-Kis strain was based on a cross between the field *An. gambiae* s.s. from Muleba District (fixed for L1014S mutation) and the laboratory susceptible *An. gambiae* s.s. Kisumu strain, resulting in a weak resistance in an out-crossed F1 generation.

To overcome the problem of low resistance, the selection of insect colonies using a sub-lethal concentration of insecticide has been extensively adopted to increase or induce heritable resistance [73,74]. Several studies have successfully induced resistance by selecting either adult mosquitoes [50,72,75,76] or larvae [50,77,78,79]. Following the insecticide selection, a pre-existing low-frequency L1014S mutation became advantageous and was selected to a higher frequency in the population. Results further indicated that out-crossing between resistant and susceptible mosquito followed by positive selection has preserved the L1014S (*kdr*-e) allele inherited from the resistant parents, as similar results were obtained in other related experiments [80]. Likewise, Song and Leu [81,82] reported the gain of rodenticide resistance alleles by susceptible house mouse *Mus musculus domesticus* through hybridization with the intrinsically resistant Algerian mouse *Mus spretus*, followed by introgression under rodenticide selection. The increased insecticide resistance and affinity to guinea pig blood observed in the Muleba-Kis strain could have been inherited via a similar mechanism and is in line with the model for the inheritance of behavioral characters in mosquitoes [83]. However, intermittent selection might be the underlying reason for the observed small rises in susceptibility of the mosquitoes, as measured by WHO susceptibility tests. This reduced resistance due to withdrawal of selection is in agreement with other studies [72]. Apart from maintaining selection for resistance, currently there is no utility for crossing the Muleba-Kis strain to field mosquitoes to maintain a complete genetic background to field populations, as the colony was established to serve as a close representative pyrethroid resistant strain, fixed for the L1014S mutation intended for phase-I and Phase-II studies. However, when the colony is intended for field release, such as in male sterile technique programs or when used to comprehend field population dynamics, it becomes even more important to renew the colony with field material to address the genetic drift and inbreeding effects [25,26].

### 4.5. Differential Resistance to Type I and Type II Pyrethroids

Pyrethroids are classified into type I and type II based on their biological responses. While type I pyrethroids result in low kill with high recovery, type II pyrethroids result in high kill with low recovery. Type I pyrethroids bind preferentially to closed channels while type II binds to open channels [84]. Research has revealed that the level of resistance in houseflies with a *super-kdr* mechanism is below 100-fold for type I and is over 200-fold for type II pyrethroids [84]. Selection of the same mosquito strain could therefore generate different resistance outcomes depending on the insecticide type, class, and concentration used, among other factors. From this study, selection of larvae with pyrethroid type I correlated with increased tolerance to type I pyrethroid papers (permethrin 0.75%) in the WHO susceptibility test, and no significant tolerance was observed against pyrethroid type II papers (alphacypermethrin, deltamethrin) following the selection. A general observation from this study indicates that type I and type II pyrethroids cause different resistance patterns, accounting for observed mosquitoes with less sensitivity to type I pyrethroids compared to type II pyrethroids. Similar results have been observed in other studies [85]. This variation is partly attributed to the different structural conformation between type I and type II pyrethroids that affect species selectivity and pyrethroid resistance [86]. Differences in structure and biological response between type I and type II pyrethroids are therefore presumed to be the underlying reasons for the different responses to selection observed in this study.

### 4.6. Metabolic Resistance

Although routine strain characterization by the WHO susceptibility test suggests that *kdr* was the underlying mechanism for resistance, limited PBO synergist bottle bioassay, which was done only once, indicated that mosquitoes’ pre-exposure to PBO results in an increased susceptibility to permethrin by 20%, suggesting the role of metabolic resistance in this strain. However, the high susceptibility of this strain to bendiocarb and pirimiphos-methyl suggests a narrow role by metabolic resistance which requires more tests to confirm its contribution to an overall resistance. There is a need for testing for the gene expression levels, especially the CYP 450 genes which have widely been linked with metabolic resistance in malaria vectors across Sub-Saharan African [87]. 

### 4.7. Intermittent Quality Control Checks and Regular Strain Authentication

In this study, the quality of the mosquito colonies was checked to ensure that the rearing and selection procedures did not lead to contamination between strains or negative effects on the mosquito’s weight or size. Underweight or undersized mosquitoes are not suitable for insecticide-testing assays, as they are more likely to be knocked down or killed by a given concentration of the insecticide. Furthermore, consistency of size is a good measure of the quality of rearing and helps to produce consistent and reproducible results provided that other rearing factors such as larval density, nutrition, environmental conditions, and microbial infection are controlled. The obtained results indicated that, despite out-crossing and insecticide selection of the strain, the weight and wing length remained fairly similar across the years, with the weight varying by only 0.001 g across four years, while wing length varied within 0.0142 mm and 0.0028 mm.

Contamination between strains held in the same facility is a regular error in mosquito rearing, especially when the same or closely related species are kept nearby [17,18,88]. The PAMVERC Test Facility keeps different strains of *An. gambiae* s.l. in different rooms and performs regular species identification using the PCR method [53] and resistance status checks to monitor for any cross-contamination. Results from characterizing the Muleba-Kisumu strain indicated that this species was identified as *An. gambiae* s.s. throughout the study, implying the absence of species contamination. *Anopheles gambiae* Muleba-Kisumu population was initially found to be partially resistant with only 30% having *kdr* fixed, but later *kdr* L1014S allele reached fixation in *A. gambiae* s.s. Muleba-Kisumu populations following the insecticide selection. These same populations exhibit strong degrees of phenotypic resistance to DDT and pyrethroid class I insecticides (permethrin). 

### 4.8. Effect of Mosquito Weight and Wing Length on Phenotypic Resistance

Data for mosquito weight from 2016 to 2020 were normally distributed. The observed deviation in 2017 in mosquito weight could partly be attributed to changes in larvae food preparation. From 2016 to 2017, the preparation of fish flakes which are used as larvae food were microwaved at 150 degrees Celsius. However, this practice was terminated in 2018 as it was suspected to increase the nutrient content of larvae food. An increase in nutrient content or food is reported to lead to longer wings [89]. Results for median weight from 2016 to 2017 when there was no selection increased; from 2017 to 2019 weight decreased significantly; then from 2019 to 2020 the selection was ongoing and mean weight remained constant. The observed increase in weight before selection was mainly due to the nutrition regimen on the larvae. On the generations from 146–158, mosquito weight was higher, with resistance thresholds equivalent to later generations (182th to 202th) when there was relatively low but maintained weight with ongoing selection. Maintaining the mosquitoes’ weight is crucial, as it is the main determinant of insecticide susceptibility, and heavier mosquitoes are more likely to survive insecticide treatment [90]. Maintaining mosquito weight from year to year is essential in getting the correct interpretation from the WHO discriminatory concentrations [90], which is fundamental in both monitoring resistance development progress and strain authentication. On the other hand, mosquito wing length results were maintained except for 2019, where they were significantly higher relative to other years. Results obtained in this study indicate that progress and status of insecticide resistance are attributed to insecticide selection and are not confounded by weight or wing length. Furthermore, in this experiment there was a detectable difference between weight and wing length, however, there were no sufficient data to prove a direct correlation between wing length and mosquito weight. Although some studies [91] have observed a correlation between weight and wing length, other studies have reported a lack of correlation between wing length and weight [54,92].

## 5. Conclusions

Since its establishment, the PAMVERC Test Facility has played an important role as a key African research player in the chain of insecticide development, particularly in screening new active ingredients for mosquito control. Successful establishment of the Muleba-Kis strain in the insectary marks an important step in the colonization of a representative East African wild *Anopheles* population characterized with *kdr*-east mutation [30]. This insectary colony enables the evaluation of vector control tools under the current East African insecticide resistance challenge. This study has also demonstrated that blood-feeding failure and low insecticide resistance in colonized mosquitoes can be overcome by out-crossing desired traits between mosquito strains followed by intermittent insecticide selection at the larval stage. It is worth mentioning that, with the interest in developing and bringing new insecticides into the market, it is crucial to quantify the fitness cost associated with resistance [93], and that although our test facility has managed to successfully create a resistant line through the described methodology with comparison to cited separate studies, further research is needed to perform a direct comparison between various selection methods and to assess the level and duration it takes to establish resistant lines in the same mosquito strain. The capacity to establish resistant mosquitoes allows for assessment of new insecticides for efficacy, cross-resistance, and the likelihood of resistance development to a novel insecticide, therefore providing an early alert to plan for an effective pre-emptive resistance management program.

## Figures and Tables

**Figure 1 insects-12-00710-f001:**
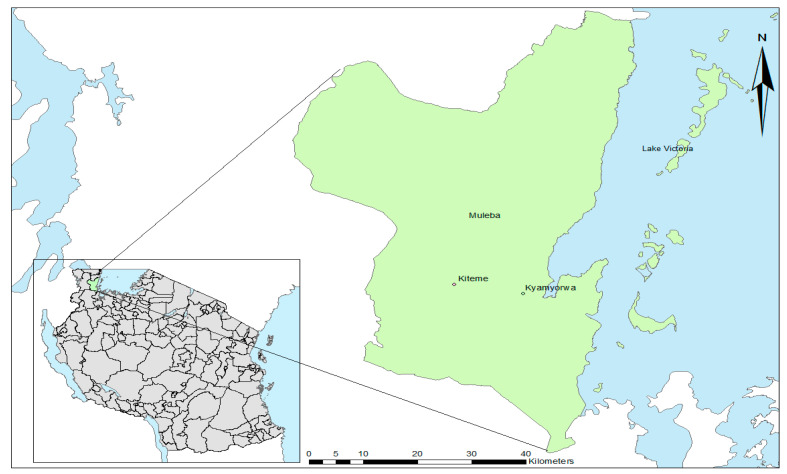
Map showing mosquito collection site in north-western Tanzania.

**Figure 2 insects-12-00710-f002:**
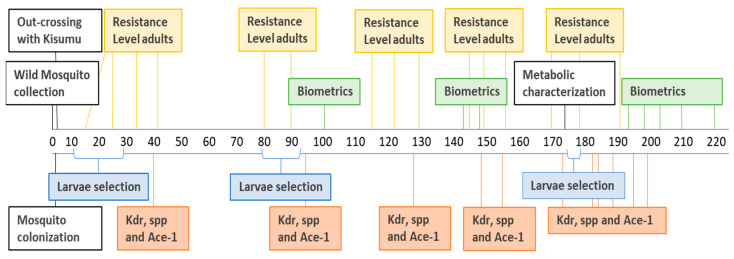
The timeline for activities, indicating wild mosquito collection, insectary colonization, out-crossing, selection, and strain characterization across *An. gambiae* Muleba-Kis generations.

**Figure 3 insects-12-00710-f003:**
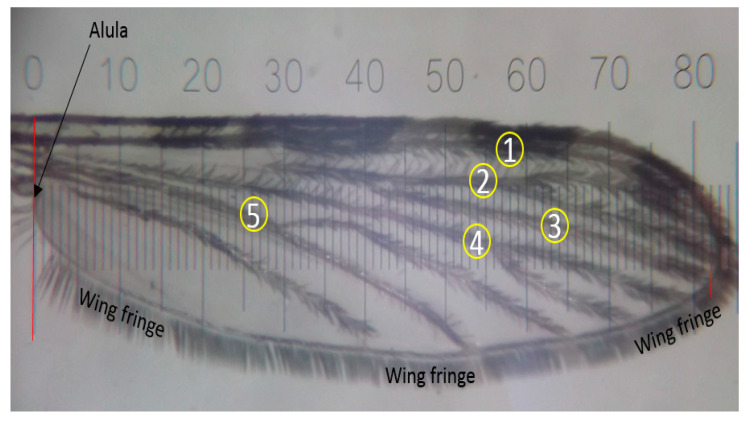
Image of wing aligned on a micrometer indicating the ocular gradations which correspond to the distance on the stage micrometer. Number 1–5 indicates the wing veins, where vein 3 is used for measuring the distance from the alula to the wing fringe (wing length). This photo was copied from PAMVERC Test Facility SOP with permission, originally taken and donated by MK (co-author).

**Figure 4 insects-12-00710-f004:**
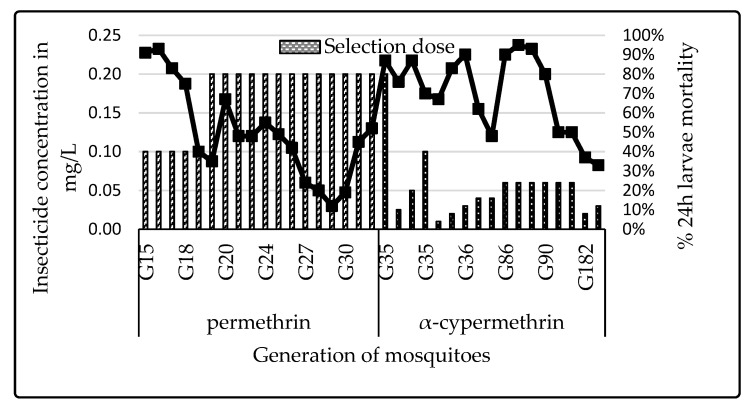
Dynamics of mortality rates of the selected larvae when different pyrethroids were used for the selection at different generations (G).

**Figure 5 insects-12-00710-f005:**
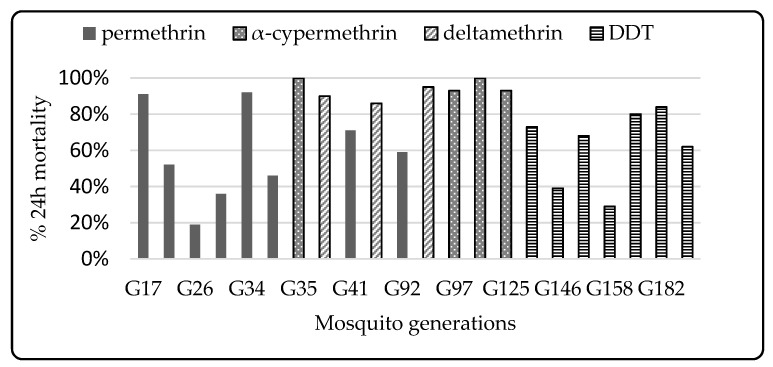
WHO susceptibility profiling of adult *An. gambiae* Muleba-Kisumu across generations. Mortality less than 90% indicates resistance, WHO (51). G = Generation.

**Figure 6 insects-12-00710-f006:**
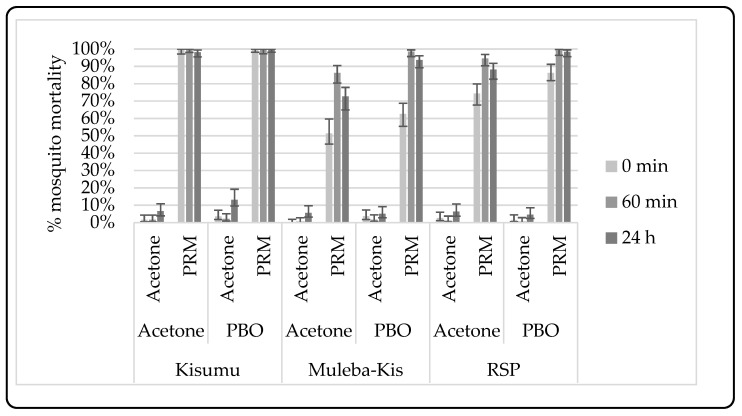
The knockdown and mortality rates of An. gambiae Kisumu and An. gambiae Muleba-Kis with and without PBO pre-exposure. Error bars are equivalent to 95% confidence intervals. PRM = Permethrin, PBO = Piperonyl butoxide.

**Figure 7 insects-12-00710-f007:**
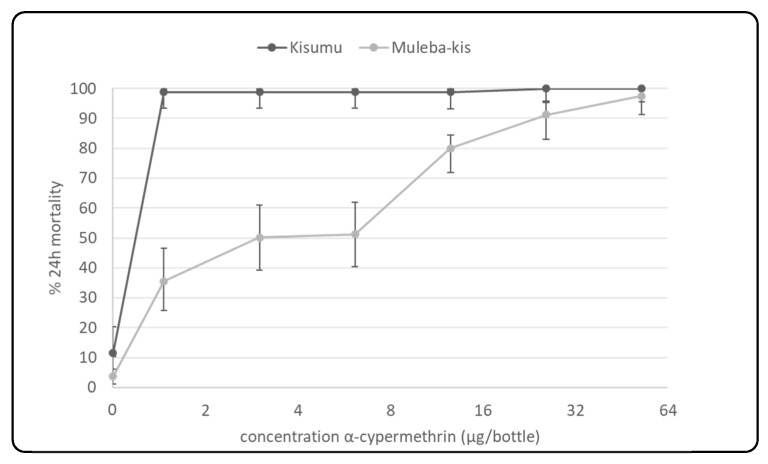
Mortality percentage of An. gambiae Kisumu and An. gambiae Muleba-Kis to varying concentrations of α-cypermethrin in CDC bottle bioassay.

**Table 1 insects-12-00710-t001:** Molecular assays for Muleba-Kis strain over generations.

Generation	Number of Samples			Molecular Assay				
		Species		*kdr*-E			*Ace*-1	
		%Ar	%Ga	%RRe	%RSe	%SSe	%RRe	%SSe
G43	37	0	100	30	27	43	0	100
G99	57	0	100	100	0	0	0	100
G131	50	0	100	100	0	0	0	100
G150	84	0	100	100	0	0	N	N
G162	100	0	100	100	0	0	N	N
G168	84	0	100	100	0	0	N	N
G178	84	0	100	100	0	0	N	N
G188	84	0	100	100	0	0	N	N
G190	84	0	100	100	0	0	0	100
G198	88	0	100	100	0	0	N	N
G204	88	0	100	100	0	0	N	N

Note: Ar = *An. arabiensis*, Ga = *An. gambiae* ss, Ace-1 = insensitive acetylcholinesterase, RRe = homozygous mutant, RSe = heterozygous mutant, SSe = homozygous susceptible. When an assay was not done it is coded as N.

**Table 2 insects-12-00710-t002:** Dynamics in mosquito wing length across years 2016, 2017, 2019, and 2020.

Year	Samples (N)	Mean Wing Length	95% CI	*p*-Value *
2016	50	2.9504	2.8995-3.0013	0.6592
2017	150	2.9362	2.9035–2.9689	
2017	150	2.9362	2.9035–2.9689	<0.0001
2019	149	3.0405	3.0066–3.0745	
2019	149	3.0405	3.0066–3.0745	0.0025
2020	100	2.9532	2.9063–3.0001	

* Two-sample T-test.

**Table 3 insects-12-00710-t003:** Dynamics in mosquito weight across years 2016, 2017, 2019, and 2020.

Year	Samples (N)	Mean Weight	95% CI	*p*-Value *
2016	50	0.0011	0.0010–0.0012	<0.0001
2017	150	0.0016	0.0015–0.0017	
2017	150	0.0016	0.0015–0.0017	<0.0001
2019	149	0.0012	0.0012–0.0013	
2019	149	0.0012	0.0012–0.0013	0.4281
2020	100	0.0012	0.0011–0.0012	

* Two-sample T-test.

## Data Availability

Datasets generated and analyzed are presented in a summarized way in this article. Full datasets will be made available from the corresponding author upon rational request.

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
