# Peer review of "Colonization and Authentication of the Pyrethroid-Resistant Anopheles gambiae s.s. Muleba-Kis Strain; an Important Test System for Laboratory Screening of New Insecticides"

_insects, 2021, doi:10.3390/insects12080710_

Round 1
Reviewer 1 Report
This study describes the procedure for introducing, colonizing, and maintaining a resistant An. gambiae s.s. strain in the laboratory. I appreciate the fact that the authors describe the challenges in ‘real life’, such as the unavailability of insecticides, and the missing of data sheets (all very familiar). Although I do believe this paper is worth publishing, I have some major concerns with the current version:
1) The introduction and discussion are well-written, but the methods and results feel like they are written in a hurry. Important details in the methods section are missing and some of the results are not adequately explained. A few examples:
- Line 223-232: split these results between larval assays and Fig 3, and adult assays and Fig 4. Now these are intertwined, and it is not immediately clear to the reader what is being discussed.
- Line 262-263, these results need a bit more explanation.
- Figure 5: what is PRM (I know it is permethrin of course)?
- Line 287: ‘against (insecticide)’ is a good example, looks like a placeholder.
- Fig 6: I suggest plotting the data on a log-scale.
- Figures 7 and 8 would be better as tables.
2) Methods:
- Did you monitor control mortality? Were tests with control mortalities >20% discarded? Did you use Abbott’s correction when needed?
- Although several mosquito mortalities are above 90%, some are still below 98%, which is suggestive of the existence of resistance. This should be pointed out.
- I suggest to show a timeline in your methods -with generations, not actual time- to indicate when which tests were performed.
- It is now impossible for me to interpret your PBO test results, as I cannot find the number of mosquitoes that were tested (nor the number of replicates).
- Same for the molecular assays: The number of mosquitoes tested are missing
3) Discussion:
- As you stress several times that you have created a mosquito line that is representative of East African mosquito populations, I expect to see more details about these natural populations (and where they are similar, but also different!) to show this statement is true.
- Where did the resistance in your line originate? The genetic variation at the start must have been extremely low (three batches of 500 eggs), which was followed by exposing low numbers (6 bowls with 100 larvae each) in each selection step. This suggests that resistance is prevalent in the local population, correct? Or have you been ‘lucky’ enough to have resistance emerge ‘de novo’ in your colony? There may be some areas/species where this process will not work, simply because background resistance is missing (or low).
3) Conclusions:
- Line 480-485: You mention on several occasions that challenges in colonizing a wild-resistant mosquito strain were overcome by your methodology. However, you do not show those failures, nor do you do a direct comparison between the two (or more) selection processes. It is true you managed to successfully create your resistant line, but you can only refer to the other methods that were unsuccessful (probably in other labs) in your discussion unless you provide data on previous attempts yourself in this paper.
- Line 485-486: This is not relevant here, and even more important: Do you not want mimic what you find in the field? If there is a high fitness cost in nature, I imagine you want to mirror this in your strain to allow for ‘fairer’ comparisons.
Minor comments:
- Line 29: ‘of the wild collection’ -> ‘of wild-collected mosquitoes’.
- Line 41: obtained in 2016, do you mean 2016-2020?
- Line 68: ‘chemistries’.
- Figure 3: ‘concentration’ instead of ‘con’.
- All figures: adjust so they are clear in B/W print.
- Line 177: Can you estimate the generation?
- Line 198: Add generations.
- Line 237: ‘methyl’ instead of ‘methy’.
- Line 253-260: Be consistent with ‘papers’ and not ‘paper’.
- Table 1: Add how many individuals were tested.
- Line 297-29: Already mentioned in methods.
- Line 299: ‘higher’ instead of ‘bigger’
- Line 355: ‘for three reasons’
- Line 432-435: How about the effect of larval densities, co-infections/disease, climate variations, or other rearing factors?
Author Response
I have provided response to all comments from the reviewer, and I have taken the suggestions into consideration for manuscript improvement

Reviewer 2 Report
I had a great preasure to review this valuable manuscript about mosquitoes' resistance to pyrethroids
Author Response
The reviewer is satisfied with the manuscript and there was no issue that was identified to be corrected.

Reviewer 3 Report
This paper describes a study with the objective of documenting and producing a robust insecticide resistant mosquito strain for future bioassays where laboratory colonisation of the species proves difficult due to blood-feeding preferences and other attributes.
The major function of the strain appears to be to gauge efficacy of a range of chemical groups against a mosquito strain that is fixed for the L1014S mutation and has a smaller component of metabolic resistance to pyrethroids, but is also fixed for ace-1. This is described in the conclusions, but would be better stated as an objective of the study in the Introduction indicating why these traits are desired.
There are also some discrepancies to be addressed to improve the paper:
I would expect the Introduction to contain more references to the literature discussing benefits and drawbacks of laboratory adaptation of insects as well as genetic requirements of an insect strain used as a comparison in bioassays. The study addresses consistency in response, maintenance of resistance and basic biological fitness (using biometrical measures only), but what about maintaining a similar genetic background to field insects, etc.? Apart from maintaining selection for resistance, will there be utility in outcrossing to field mosquitoes at particular intervals?
It would also be useful if the Introduction described the effect in the study species of L1014S in terms of comparative resistance to Type I and Type II pyrethroids and DDT, if relevant. There is no information about frequency of the two target-site mutations prior to G49. Is any information available from earlier generations or the parental populations?
Table 1. Column %RRe is repeated.
Line 245: you describe a steady decrease in susceptibility to permethrin from the 17th to the 35th generation (shown in Figure 3). In fact, the lowest mortality is shown to occur at G29 (~10%), after which it starts to increase again and is ~50% at G35. Please explain/correct this statement.
Line 260: again in reference to Figure 3, deltamethrin is said to be below the "cut-off value", but at G35 and G89, this is not strictly correct. Please make an accurate description of the results.
Figure captions must stand alone. Please include genus and species in the caption for each relevant figure. For Figure 5, please include a definition of P, R and M in the caption.
Bars in Figures 3 and 6 show no confidence intervals.
I think the paper would benefit from a diagram describing the initial crossing scheme and perceived attributes brought into the strain, along with a further timeline of resistance selection events and other outcrosses or quality control activities required to maintain the strain.
This statement appears in your manuscript, yet guinea pigs were used to feed mosquitoes. 'Institutional Review Board Statement: Not applicable. This study did not involve humans or animals.' Please explain and/or correct this statement.
References - genus and species' names have not been italicised, there is some unnecessary capitalisation in some of the titles, volume and page numbers are missing in many cases.
Author Response
I have worked on the comments and suggestions that were given by the Reviewer. All action points have been responded point-by-point. Please see attached responses

Round 2
Reviewer 3 Report
The response to the initial review has satisfied most of my queries and the manuscript is improved. There are a few minor editorial points to be addressed:
Line 72: change 'undergoes' to 'undergo'
Line 93: change to: 'representative for a few'
Line 107: change to: 'has been maintained'
Line 122: 'Reiner' should be 'Reimer'
Line 123: please check and clarify this sentence. I think it should be ...populations carrying a high kdr frequency showed more resistance to DDT and Type I pyrethroids than to Type II pyrethroids.
Line 136: change to: 'to resemble the parental resistant population'
Line 155: change to: 'were the main vectors'
Line 160: species name should be in italics
Line 165: change to: 'house bedrooms'
Line 166: change to 'glucose solution' and indicate % used.
Line 177: change 'Too' to 'to'
Line 181 and 187 - spelling of vigour is inconsistent
Line 183: change to 'one hour prior to blood-feeding'
Line 193: use degree sign after 35
Line 226: change to 'Each type of insecticide bioassay was performed'
Line 233: why are there three decimal places for 12.556?
Line 239: there is an extra space before the comma.
Line 246 and 248: change 'mosquitos' to 'mosquitoes'
Line 296: change to 'selection was infrequent'
Line 304: change to 'decreased to less' and 'follows a similar trend'
Line 319: change to 'were considered for analysis.'
Line 370: change to: only significantly higher'
Line 450: change to 'programs'
Line 485: I do not understand the reference to 'mosquito's species composition'
Line 507: change to 'were normally distributed.'
Line 544: change to: 'test facility has managed'
Line 561. There two full stops here.
Throughout the document the term 'kdr' is sometimes in italics and sometimes not.
There should be a space between numbers and their units.
The references still need further improvements in formatting.
Author Response
I have made corrections for all the areas pointed out by the Reviewer
